# In Silico Screening of Bioactive Compounds of Representative Seaweeds to Inhibit SARS-CoV-2 ACE2-Bound Omicron B.1.1.529 Spike Protein Trimer

**DOI:** 10.3390/md20020148

**Published:** 2022-02-17

**Authors:** Muruganantham Bharathi, Bhagavathi Sundaram Sivamaruthi, Periyanaina Kesika, Subramanian Thangaleela, Chaiyavat Chaiyasut

**Affiliations:** 1Innovation Center for Holistic Health, Nutraceuticals, and Cosmeceuticals, Faculty of Pharmacy, Chiang Mai University, Chiang Mai 50200, Thailand; bharathi.m03@gmail.com (M.B.); sivamaruthi.b@cmu.ac.th (B.S.S.); drthangaleela@gmail.com (S.T.); 2Office of Research Administration, Chiang Mai University, Chiang Mai 50200, Thailand

**Keywords:** SARS-CoV-2, omicron B.1.1529, seaweeds, *Sargassum wightii*, *Corallina officinalis*

## Abstract

Omicron is an emerging SARS-CoV-2 variant, evolved from the Indian delta variant B.1.617.2, which is currently infecting worldwide. The spike glycoprotein, an important molecule in the pathogenesis and transmissions of SARS-CoV-2 variants, especially omicron B.1.1.529, shows 37 mutations distributed over the trimeric protein domains. Notably, fifteen of these mutations reside in the receptor-binding domain of the spike glycoprotein, which may alter transmissibility and infectivity. Additionally, the omicron spike evades neutralization more efficiently than the delta spike. Most of the therapeutic antibodies are ineffective against the omicron variant, and double immunization with BioNTech-Pfizer (BNT162b2) might not adequately protect against severe disease induced by omicron B.1.1.529. So far, no efficient antiviral drugs are available against omicron. The present study identified the promising inhibitors from seaweed’s bioactive compounds to inhibit the omicron variant B.1.1.529. We have also compared the seaweed’s compounds with the standard drugs ceftriaxone and cefuroxime, which were suggested as beneficial antiviral drugs in COVID-19 treatment. Our molecular docking analysis revealed that caffeic acid hexoside (−6.4 kcal/mol; RMSD = 2.382 Å) and phloretin (−6.3 kcal/mol; RMSD = 0.061 Å) from *Sargassum wightii* (*S. wightii*) showed the inhibitory effect against the crucial residues ASN417, SER496, TYR501, and HIS505, which are supported for the inviolable omicron and angiotensin-converting enzyme II (ACE2) receptor interaction. Cholestan-3-ol, 2-methylene-, (3beta, 5 alpha) (CMBA) (−6.0 kcal/mol; RMSD = 3.074 Å) from *Corallina officinalis* (*C. officinalis*) manifested the strong inhibitory effect against the omicron RBD mutated residues LEU452 and ALA484, was magnificently observed as the essential residues in Indian delta variant B.1.617.2 previously. The standard drugs (ceftriaxone and cefuroxime) showed no or less inhibitory effect against RBD of omicron B.1.1.529. The present study also emphasized the pharmacological properties of the considered chemical compounds. The results could be used to develop potent seaweed-based antiviral drugs and/or dietary supplements to treat omicron B.1.1529-infected patients.

## 1. Introduction

Omicron B.1.1.529 variant was first reported to World Health Organization (WHO) from South Africa on 24 November 2021 [1]. In South Africa, the epidemiological circumstance has been characterized by three distinct peaks in confirmed incidents, the most recent predominated by the Indian Delta variant B.1.617.2 [2]. On 26 November 2021, the World Health Organization’s Technical Advisory Group on Virus Evolution (TAG-VE) proposed that variant B.1.1.529, referred to as omicron, be designated as a variant of concerns (VOCs) [1]. Infections have increased dramatically in recent weeks, coinciding with the global detection of the omicron B.1.1.529 variant. Globally, the number of countries registering SARS-CoV-2 omicron B.1.1.529 infections keeps climbing, with 117 countries reporting 354 million confirmed cases as of 15 January 2022 [3]. Due to the widely altered omicron variant, many nations have imposed travel restrictions or border closures, and extensive research is desperately required for medical examinations and border closure advantages [4].

In the receptor-binding domain (RBD), the crucial mutations G446S, N501Y, G496S, Y505H, Q493R, E484A, T478K, S477N, K417N, G339D, S371L, S373P, S375F, N440K, and Q498R occur in the RBD of omicron B.1.1.529 variant [5,6,7]. The mutated residues Q493R, G496S, and Q498R, in particular, strengthen RBD adherence to the angiotensin-converting enzyme 2 (ACE2) receptor [5,8]. Additionally, the N501Y mutation localizes at RBD, which might help attain higher binding to host cells and result in elevated transmission and infectiousness [9]. Similarly, the mutation in E484K, K417N/K417T, and N501Y plays a vital role in the RBD and the ACE2 receptor binding of the beta (B.1.351) and gamma (P.1) variants. Therefore, B.1.351 and P.1 were designated as VOCs in early 2021, an outbreak in South Africa and Brazil [10,11,12].

Omicron B.1.1.529 has many nonsynonymous mutations in the RBD, including in L452R, and E484K, which significantly neutralize the activity of multiple monoclonal antibodies (mAbs) [13,14]. Omicron may be more contagious than any other identified variant, and more than twice as infectious as the delta variant B.1.617.2, possibly due to the mutations in N440K, T478K, and N501Y [15,16]. Moreover, omicron has the potential to disrupt the binding of most 185 mAbs, indicating more vaccine-breakthrough potential than delta or any other identified variations [17].

The world demands innovative antiviral medications that are efficient at halting viral propagation without exerting further selection pressure on resistant variants [18]. Aside from upgrading vaccinations against omicron or future variants, we must accelerate the discovery of novel antivirals [19].

Seaweeds are important resources from coastal ecosystems, contributing considerably to the trophic structure [20,21]. Furthermore, seaweeds provide several bioactive compounds with antiviral, anticancer, and anticoagulant activity and help manage conditions like obesity and diabetes [22,23,24].

One of the main processes involved in viral proliferation and subsequent diseases is the overproduction of reactive oxygen species (ROS) and the depletion of antioxidant systems [25,26]. The sulfated polysaccharides derived from *Corallina officinalis* (*C. officinalis*) would be used as a natural antioxidant in the food manufacturing industry [27,28]. *Caulerpa racemosa* (*C. racemosa*) could be a rich source of novel antioxidant and anti-inflammatory agents. However, the bioactive chemicals must be separated and purified to study the exact mechanism of action [29,30]. Furthermore, increasing soluble Fas and B-cell CLL/lymphoma-2 (Bcl2) levels in COVID-19 patients raises the fatality rate in COVID-19 individuals [31]. However, *Colpomenia sinuosa* (*C. sinuosa*)-treated HCT-116 cells have downregulated anti-apoptotic Bcl 2, which leads to caspases activation [32].

*Gracilaria edulis* (*G. edulis*) and *Gracilaria corticata* (*G. corticata*) possess greater levels of essential n-3 polyunsaturated fatty acid (PUFA), which inhibits the establishment of atherosclerotic plaque, blood coagulation, and blood pressure, as well as boost immunological function [33,34]. 

*Padina boergesenii* (*P. boergesenii*) might contain bioactive chemicals with substantial antibiotic action and play an important role in drug discovery [35]. 

*Sargassum wightii* (*S. wightii*) is a brown seaweed containing several bioactive compounds such as fucoidan and fucoxanthin, and substantial quantities of polyphenols and flavonoids play a vital role in anticoagulant, antithrombotic, immunomodulation, anticancer, and antiviral activity [36,37]. 

The phytocompounds of *C. officinalis*, *C. racemosa*, *C. sinuosa*, *G. edulis*, *G. corticata*, *P. boergesenii*, *S. wightii*, and *U. fasciata* were used in this study to find potent inhibitors of SARS-CoV-2 ACE2-bound omicron B.1.1.529 spike protein trimer. We conducted the molecular docking study using Autodock Vina [38] with Pyrx v0.8 [39], Pymol v2.5 [40], Ligplot+ v2.2.4 [41], and Discovery Studio Visualizer v21.1.0.20298. We also assessed the drug-likeness, adsorption, digestion, metabolism, excretion, toxicity (ADMET), toxicity class, and a lethal dosage study for the finalized chemical compounds using the Molinspiration server [42], ADMETlab 2.0 [43], and ProTox-II [44].

## 2. Materials and Methods

### 2.1. Chemical Compounds from Seaweeds

A sum of 96 chemical compounds from seven seaweeds (*C. officinalis*, *C. racemosa*, *C. sinuosa*, *G. edulis*, *G. corticata*, *P. boergesenii*, *S. wightii*, and *U. fasciata*) was subjected to evaluate the interaction with the RBD of S-protein of omicron variant B.1.1.529.

### 2.2. Target Preparation and Ligand Library

Cryo-EM structure of the ACE2 binding, spike protein of SARS-CoV-2 omicron spike PDB: 7T9J [5], was retrieved from protein data bank and exported in Protein Data Bank (PDB) format [45]. The major chemical compounds of the selected seaweeds were retrieved in Spatial Data File (SDF) format and 2D structures from the PubChem database [46]. For the acquired 2D structures, the 3D structure was delineated and optimized with the force field relying on Chemistry at Harvard Macromolecular Mechanics (CHARMM) approximation using ACD/Chemsketch vC05E41 (Advanced Chemistry Development, Inc., Toronto, Canada) and saved as SDF formats. The 3D structure of all chemical compounds was then transformed to the PDB format for the further docking process using the OPEN BABEL program [47].

### 2.3. Molecular Docking

After preparing the chemical compounds as ligands, and RBD from 7T9J as a target, we implemented the Autodock Vina for the molecular docking process using PyRx software [38,39]. It examines the docking propensity of the ligands and the interfaces between the ligands and the RBD residues. The prepared targets and ligands were saved in PDBQT format. We used the PyRx virtual screening tool for the docking analysis to identify the crucial ligands, which inhibit the ACE2 binding residues. The size was assigned to the grid box properties as size_x = 67.33 Å, y = 74.87 Å, and size_z = 106.22 Å, and the ligands were docked with the RBD. The ligands were chosen based on their binding affinity (≤6.0 kcal/mol.) by importing the docked data into PyMol [40], Lig-Plot^+^ [41] and Discovery Studio Visualizer v19.1.0.1828 (Dassault Systèmes BIOVIA, Rue Marcel Dassault, Vélizy-Villacoublay-78140, France, www.accelerys.com (accessed on 6 January 2022)), and the substantial association for both the ligands and the receptors’ binding site was obtained in 2D and 3D format. Auto dock vina scoring function was employed.
$$C=\underset{i<j}{\sum }f_{titj\;\left(r_{ij}\right)}$$
where *C* is the sum of intermolecular and intramolecular distances; ∑ is atoms separated by three consecutive covalent bonds; *ftitj* is the symmetric set of interaction functions; *r_ij_* is interatomic distance.

### 2.4. Evaluation of Ligands Drug-Likeness and Toxicity

Using the Molinspiration server (www.molinspiration.com/cgi-bin/properties (accessed on 15 January 2022)), the finished ligands were tested for drugability, physicochemical characteristics, toxicity, toxicity classes, and fatal dosage. The drugability characteristics were evaluated using molinspiration lipophilicity (Mi log P), molar weights (MW), total polar surface area (TPSA), number of rotatable atoms (natoms), hydrogen bond acceptor (HBA), and hydrogen bond donor (HBD). Lipinski’s rule of the drug-like compounds was determined. In addition, the PubChem Database [46] was used to download the simplified molecular-input line-entry system (SMILES) to compute ADMET characteristics with toxicity class. The ADMET properties were calculated using ADMETlab 2.0 [43] and ProTox-II with default parameters [44].

## 3. Results

The cryo-EM structure of the ACE2 binding RBD of omicron B.1.1.529 spike protein (PDB: 7T9J) with the RMSD of 2.79 Å [5] is depicted with its mutation in Figure 1.

The effects of chemical compounds from *C. officinalis*, *C. racemosa*, *C. sinuosa*, *G. edulis*, *G. corticata*, *P. boergesenii*, *S. wightii*, and *U. fasciata* were analyzed to understand the binding efficiency against the omicron spike protein of RBD (Appendix A; Supplementary File S1). The chemical properties, including molecular formula, molecular weight, and PubChem ID for the seaweed compounds, which showed significant activity against the RBD of omicron spike protein (binding affinity ≤ −6.0 kcal/mol) were listed (Table 1), and its Ligplot interactions were analyzed and represented (Appendix A). 

The phytocompounds interacting with the RBD residues of omicron spike protein, binding affinity, RMSD, and its interacting residues are listed in Table 2. The chemical compounds based on the crucial residue interactions such as PRO373, PHE375, TYR396, ASN417, LYS440, LEU452, ALA484, SER496, TYR501, and HIS505 and the binding affinity ≤ −6.0 kcal/mol were analyzed for the ADEMT properties such as drug-likeness, lethal dose, and toxicity classes and are represented in Table 3.

The standard drug, cefuroxime, evidenced the hydrogen bonding interactions with THR376, ARG408, and TYR508, and interacted with ARG408, VAL503 through the pi-alkyl bond. It also interacted with the residue GLY404 using a carbon–hydrogen bond and was surrounded by hydrophobic residues such as PHE375 and ASN437 with the binding affinity of −5.3 kcal/mol and RMSD of 2.483 Å. Furthermore, cefuroxime interacted with one crucial mutated residue of RBD, PHE375, with a lower binding affinity (Figure 2A–C). Additionally, cefuroxime toxicity classes were predicted as VI. The 3’,8,8’-Trimethoxy-3-piperidin-1-yl2,2’-binaphthyl-1,1’,4,4’-tetrone (TPBT) from *C. officinalis* formed the pi-sigma and pi-cation interaction with PHE464 and GLU516, a pi-alkyl interaction with PRO426 and PRO463, and a hydrophobic interaction with ARG355, TYR396, ASP428, PHE429, THR430, SER514, PHE515, LEU517, and LEU518 residues with the binding affinity of −6.9 kcal/mol and RMSD of 1.897 Å. Likewise, the toxicity class was predicted as IV due to its moderate carcinogenic, immunogenic, mutagenic, and cytotoxic effects (Figure 2D–F). Similarly, cholestan-3-ol, 2-methylene-, (3 beta, 5 alpha) interacted with SER494 and PHE490 through hydrogen and pi-sigma bond and also interacted with the residues LEU452, TYR449, ILE472, ALA484, and PHE490 through alkyl interaction, surrounded by the hydrophobic residues GLY482 and THR470 with the binding affinity of −6.0 kcal/mol, and RMSD of 3.074Å. The toxicity class was predicted as V because of its positive effect on immunogenicity (Figure 2G–I).

Syringaresinol from *P. boergesenii* interacted with ARG355, ASP427 *, PRO463 *, PHE515 *, and GLU516 through the hydrogen and carbon–hydrogen bond with the binding affinity of −6.4 kcal/mol and RMSD of 0.44Å. It also extended its pi-sigma and pi-alkyl interaction with PHE464, TYR396, PRO426, LYS462, and PRO463. Additionally, syringaresinol was surrounded by hydrophobic residues such as ASP428, THR430, and SER514. The toxicity class was predicted as IV, described by moderate carcinogenicity and a positive immunotoxic effect (Figure 2J–L).

Caffeic acid hexoside from *S. wightii* interacted with ARG403, GLU406, ASN417, TYR453, and SER496 via hydrogen bond with the binding affinity of −6.4 kcal/mol and RMSD of 2.82Å. It was also surrounded by hydrophobic residues such as ASP405, ARG408, GLN409, ILE418, LEU455, TYR495, SER494, TYR501, and HIS505. The toxicity class was identified as V, which also showed a positive effect on immunotoxicity (Figure 2M–O). Phloretin had a hydrogen bond interaction with TYR501, SER496, and TYR453 with the binding affinity of −6.3 kcal/mol and RMSD of 0.061Å. It also exhibited the interaction with the residues TYR501 and HIS505 via pi-pi-stacked bond, surrounded by hydrophobic residues such as ARG403, TYR495 PHE497, THR500, and GLY502. The predicted toxicity class for phloretin was IV, which indicated moderate hepatotoxic and mild carcinogenic effects (Figure 2P–R).

The Ligplot interactions for cefuroxime, TPBT, cholestan-3-ol, 2-methylene-, (3 beta, 5 alpha), syringaresinol, caffeic acid hexoside, and phloretin with the RBD were interpreted and are depicted in Figure 3A–F. The standard drug, ceftriaxone, interacted with ARG403, ASN417, TYR453, SER494, SER496, and TYR501 by hydrogen bond and formed the pi-alkyl interactions with ARG403 and HIS505. It was surrounded by hydrophobic residues such as ASP405, GLU406, ARG408, GLN409, LEU455, and ARG49 with the binding affinity of –7.1 kcal/mol, and RMSD of 43.189Å (Table 2). However, according to the observed RMSD (43.189Å), ceftriaxone did not interact with the RBD of omicron B.1.1.529 spike protein.

In addition, the Ligplot interactions between the RBD of omicron B.1.1.529 spike protein and the selected compounds such as cefuroxime, TPBT, CMBA, syringaresinol, caffeic acid hexoside, and phloretin are depicted in Figure 3.

## 4. Discussion

Natural compounds have been utilized internationally for COVID 19 owing to their chemical variety, species diversity, and drug-like properties [48,49,50]. The recent experimental studies described that a lack of micro- and macronutrients in terms of quality and/or quantity in the diet of SARS-CoV-2-infected patients increases the morbidity and mortality rates [51]. Nevertheless, the intake of seaweeds as dietary supplements may provide the essential micronutrients, reducing the risk of omicron infections and organ deterioration [52,53]. The chemical compounds from the dietary seaweeds, particularly constituted with fucoidan, might have antiviral actions against SARS-CoV-2 [54]. Furthermore, seaweeds are effective against a wide spectrum of viruses such as the human immunodeficiency virus, herpes simplex, vesicular stomatitis, and cytomegalovirus [55]. Hence, in the current study, the seaweed compounds were evaluated for their antiviral activity against the binding of omicron B.1.1.529 with the ACE2 receptor.

Omicron B.1.1.529 variant had 37 mutations in the spike protein [56]. Among that, 17 mutations such as Leu371, Pro373, Phe375, Tyr396, Asp339, Lys440, Trp353, Asn417, His505, Tyr501, Arg493, Ser496, Arg498, Ser446, Ala484, Lys478, and Asn477 are reported in the RBD (Figure 1). These amino acid substitutions might have a major impact on transmission and evasion of neutralizing mAbs, including BNT162b2, PiCoVacc, and AZD1222 vaccines [57,58].

Cefuroxime showed potential activity against protease, RNA-dependent RNA polymerase protein, and ACE2-Spike complex, which might be therapeutic ailments to fight against SARS-CoV-2 [59].

In the present study, cefuroxime exhibited a lower binding affinity (−5.3 kcal/mol), and interacted with PHE375 in RBD of omicron B.1.1.529 spike protein (Figure 2A–C, Table 2). Ceftriaxone established the interaction with omicron, even though the binding affinity was −7.1 kcal/mol and RMSD was 43.189 Å. Ceftriaxone was recommended to treat SARS-CoV-2-infected patients before ICU admission [60,61].

Bicyclo[3.2.1]oct-3-en-2-one, 3,8-dihydroxy-1-methoxy-7-(7-methoxy-1,3-benzodioxol-5-yl)-6-methyl-5 (BDBM) of *G. corticata* showed a binding affinity of −6.3 kcal/mol, and RMSD of 20.322Å (Table 2). Similarly, glycitein, naringenin, and pyrano [4,3-b] benzopyran-1,9-dione, 5a-methoxy-9a-methyl-3-(1-propenyl) perhydro from *P. boergesenii* showed binding affinities of −6.1, −6.4, and −6.1 kcal/mol with RMSD of 11.668, 20.209, and 29.617Å, respectively, to the RBD of omicron spike protein (Table 2). The increased RMSD values (thresholds ≤ 3.0Å) indicated that the compounds were “unfit” to bind with the RBD of the omicron spike protein since the native structure resolution was specified as 2.79Å.

Glycitein and naringenin were analyzed against the spike protein of SARS-CoV-2 and classified as antiviral flavonoids used to treat SARS-CoV-2-affected patients [62,63]. Even though the current investigation demonstrated that ceftriaxone, BDBM, glycitein, and naringenin were not really bound to the RBD of the omicron spike protein and would not be useful for the development of antiviral medications against omicron B.1.1.529. whereas cholesta-8,24-dien-3-ol, 4-methyl-, (3 beta, 4 alpha)- from *G. corticata* exhibited −6.8 kcal/mol of binding affinity and 1.544 Å of RMSD towards the RBD, which might be useful to treat the omicron-affected patients (Table 2).

Omicron RBD mutant tightly interacts with ACE2 compared to the delta variant due to the T478K and L452R mutation [64]. Accordingly, the compound TPBT from *C. officinalis* significantly acted against the RBD of omicron spike protein, especially inhibiting the mutated residues such as TYR396 (Figure 2D–F) (Table 2). Kulkarni et al. report that the CMBA shows better binding affinity to the RBD of SARS-CoV-2 spike protein without hydrogen bonding interactions [65]. In contrast, CMBA (−6.0 kcal/mol) from *C. officinalis* substantially fiddled against the RBD of omicron spike protein, especially inhibited the mutated residues L452R and E484A. Additionally, the hydrogen bond formed with the SER494 residue (Figure 2G–I) (Table 2) revealed that the CMBA compound could be strongly recommended as an antiviral drug-like compound against omicron B.1.1.529 to treat omicron-infected patients.

Adem et al. reported that caffeic acid derivatives have potential antiviral effects which also act substantially against the SARS-CoV-2 [66]. Caffeic acids have been reported for their potent virucidal activity against the herpes simplex virus, severe fever with thrombocytopenia syndrome virus, and influenza virus [67,68,69]. In our study, caffeic acid hexoside (−6.4 kcal/mol) from *S. wightii* inhibited ASN417, SER496, TYR501, and HIS505. These residues were reported as the strongest ACE2 receptor-binding residues in the RBD of omicron spike protein (Figure 2M–O) (Table 2). The current study results distinctly demonstrated that caffeic acid hexoside had a substantial inhibitory effect against omicron B.1.1.529.

Muhammad et al. conclude that antioxidant levels are depleted due to their increased utilization in counterbalancing free radicals’ negative effect. Meanwhile, the levels of erythrocytes glutathione (GSH), glutathione peroxidase (GPx), catalase, and plasma superoxide dismutase (SOD) are observed to be lower in COVID-19 patients when compared with those of controls [70]. GPx and SOD are essential antioxidant enzymes that scavenge free radicals, especially superoxide anion radicals [71]. The reduced level of GPx and SOD leads the oxidative stress and inflammation in SARS-CoV-2-affected patients [72]. Fortunately, phloretin can activate transcription factors that induce antioxidant genes’ expression, improve the enzymatic antioxidant defense system, and increase SOD and GPx concentration [73]. Additionally, phloretin can be utilized as a penetration enhancer of administered drugs due to its increased fluidity towards the binding of biological membranes. Phloretin (−6.3 kcal/mol) from *S. wightii* inhibited SER496, TYR501, and HIS505, which were reported as the strongest ACE2 receptor-binding residues in the RBD of omicron spike protein (Figure 2P–R) (Table 2). They suggested that the development of *S. wightii*-based dietary supplements and caffeic acid hexoside and phloretin-based antiviral medications against omicron B.1.1.529 could be feasible to manage the infection.

Glucobrassicin was reported to have inhibitory effects against SARS-CoV-2 protease [74]. Similarly, glucobrassicin from *C. racemosa* and *P.boergesenii* demonstrated the significant inhibition against RBD spike protein with the binding affinity of −6.8kcal/mol, even though it did not interact with the crucial residues that bonded with the ACE2 receptor (Table 2). Glucobrassicin exhibited strong hydrogen bonding interactions with ARG457, ARG466, and ASP467 residues and may be considered for development of antiviral drugs to control the infection rate and organ damage caused by omicron B.1.1.529 (Table 2). However, additional experimental studies are needed to understand glucobrassicin’s inhibitory mechanism against the omicron B.1.1.529.

A high-quality drug candidate should be effective against the therapeutic target and should also exhibit adequate ADMET qualities at therapeutic dosage to avoid drug failure in the clinical phases [75]. Hence, in this current study, we implemented a series of in silico tools to predict the ADMET properties (absorption, distribution, metabolism, excretion, and toxicity) and toxicity classes based on hepatotoxicity, carcinogenicity, immunotoxicity, mutagenicity, cytotoxicity, and the lethal dose for the resulted compounds. The classified toxicity such as hepatotoxicity, carcinogenicity, immunotoxicity, mutagenicity, and cytotoxicity might lead to liver damage, tumor formation, allergic reaction, and DNA damage. However, the functional group can be substituted by the other group of elements using the lead optimization method, which is described as a crucial part of the drug development process [76,77].

Cefuroxime possesses more than 10 hydrogen bond acceptors and exhibits mild to moderate hepatotoxicity and cytotoxicity (Table 3). Nevertheless, cefuroxime hydrophobically bonded with the crucial residue PHE375 with a reduced binding affinity (Table 2 and Table 3) and showed a lesser inhibitory effect against omicron B.1.1.529.

TPBT was perfectly satisfied with drug-likeness standards (“Rule of Five” of Lipinski et al. [78]) and was found to be a more effective antiviral molecule. TPBT was bonded with TYR396 residue, which is essential for the ACE2 receptor attachment. Even though the level of toxicity was high, it was a hepatic-friendly drug that would be useful to treat omicron B.1.1.529-affected patients (Table 2 and Table 3).

Similarly, the lipophilicity of the CMBA was observed as logP 8.11, which would increase the binding affinity with LEU452 and ALA484 of omicron spike protein [79,80] (Table 2 and Table 3). CMBA was observed as a hepatic-friendly and a noncytotoxic candidate that would be an effective drug against omicron B.1.1.529.

Cholesta-8,24-dien-3-ol, 4-methyl-, (3 beta, 4 alpha) lipophilicity was increased, which might be the reason for the binding with TYR396 [81] (Table 2 and Table 3). Like TPBT, syringaresinol also completely obeyed Lipinski’s rule and bonded with the crucial residue TYR396 and was also observed as a hepatic-friendly drug with reduced cytotoxicity, which might as well be used as an antiviral drug candidate to inhibit omicron infections rate (Table 2 and Table 3).

Caffeic acid hexoside contains more than five hydrogen donors and bonded with the most crucial residues of the RBD domain, such as ASN417, SER496, TYR501, and HIS505 (Table 2 and Table 3). However, natural products with the violation of two or more Lipinski rules are acceptable [82]. It revealed that caffeic acid hexoside might be considered an antiviral candidate molecule, specifically against omicron B.1.1.529.

In addition, phloretin was found to be a drug-like compound [73]. Phloretin bonded with crucial residues such as SER496, TYR501, and HIS505 of RBD, which are responsible for the binding of ACE2 receptors. Thus, phloretin would be suggested as the antiviral drug against omicron B.1.1.529 (Table 2 and Table 3).

In conclusion, CMBA, caffeic acid hexoside, and phloretin are the most effective drug-like compounds against the omicron B.1.1.529 spike protein while considering the ADMET properties, drug-likeness, and the number of crucial residues such as S375F, TYR396, K417N, L452R, E484A, G496S, N501Y, and Y505H. These residues are reported as the primary cause for the strong binding of omicron B.1.1.529 with the ACE2 receptor [5]. Thus, seaweeds might be used as an important alternative dietary supplement and a promising candidate for developing antiviral drugs to treat and manage the omicron B.1.1.529 infection. However, further experiments are required to sustain the findings of the current study.

## 5. Conclusions

The present study primarily describes seaweed compounds for the inhibition of binding of RBD of omicron B.1.1.529 spike protein with the ACE2 receptor. Cholestan-3-ol, 2-methylene-, (3 beta, 5 alpha) from *C. officinalis* acted against the LEU452 and ALA484 residues of the omicron B.1.1.529 spike protein. Similarly, caffeic acid hexoside and phloretin from *S. wightii* inhibitedACE2-omicron spike protein-binding residues, specifically ASN417, SER496, TYR501, and HIS505. Further experimental studies are needed to endorse the finding of the current study. Nevertheless, our findings could help medical practitioners and dieticians to treat and manage the omicron B.1.1.529 infection. Additionally, the present study provided a strong foundation to develop antiviral drugs and/or novel food supplements from *C. officinalis* and *S. wightii*.

## Figures and Tables

**Figure 1 marinedrugs-20-00148-f001:**
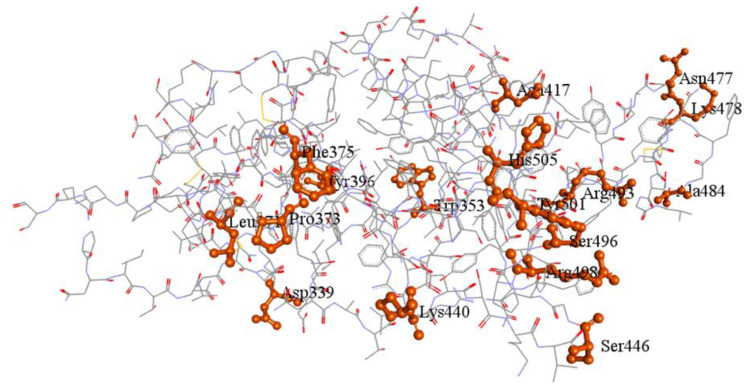
The cryo-EM structure of the RBD of spike protein of SARS-CoV-2 omicron B.1.1.529 (PDB: 7T9J). The critically mutated regions are highlighted in orange.

**Figure 2 marinedrugs-20-00148-f002:**
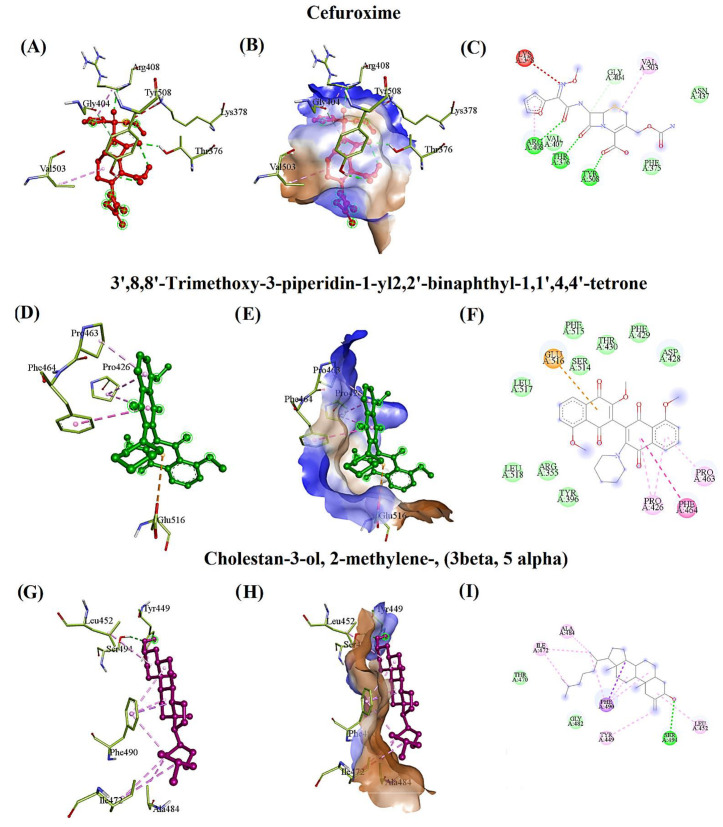
The docking pose of the RBD of omicron B.1.1.529 spike protein with the most promising phytocompounds based on the binding affinity and interacting residues (**A**,**D**,**G**,**J**,**M**,**P**). The hydrophobic surface of RBD with standard drugs and seaweeds compounds (**B**,**E**,**H**,**K**,**N**,**Q**). The hydrophobicity of the interacting residues (brown (↑ hydrophobicity)-blue (↓ hydrophobicity). The type of bonds involved in interacting phytocompounds with RBD residues (**C**,**F**,**I**,**L**,**O**,**R**).

**Figure 3 marinedrugs-20-00148-f003:**
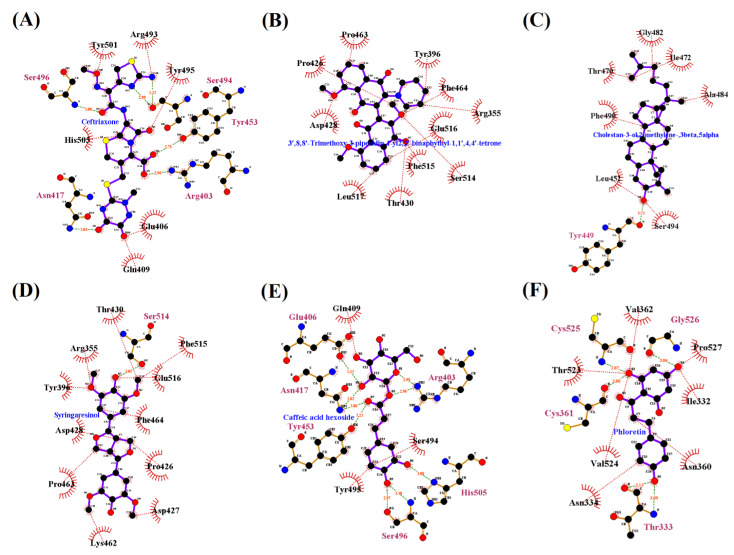
The Ligplot interaction of the RBD of omicron B.1.1.529 spike protein with the most promising chemical compounds and standard drugs (**A**–**F**).

**Table 1 marinedrugs-20-00148-t001:** The properties of screened phytocompounds with the binding energy of ≤−6.0 kcal/mol against RBD domain of omicron B.1.1.529 spike protein.

Seaweeds Name	Compound Name	Molecular Formula	Mol. Weight (g/mol)	PubChem ID
Standard drug	Ceftriaxone	C_18_H_18_N_8_O7S_3_	554.6	5479530
	Cefuroxime	C_16_H_16_N_4_O_8_S	424.4	5479529
*C. officinalis*	3’,8,8’-Trimethoxy-3-piperidin-1-yl2,2’-binaphthyl-1,1’,4,4’-tetrone	C_28_H_25_NO_7_	487.5	590815
	Cholestan-3-ol, 2-methylene-, (3beta, 5 alpha)	C_28_H_48_O	400.7	22213932
*C.racemosa*	Glucobrassicin	C_16_H_20_N_2_O_9_S_2_	448.5	656506
*C. sativa L*	Matairesinol	C_20_H_22_O_6_	358.4	119205
	Naringenin	C_15_H_12_O_5_	272.25	932
	Syringaresinol	C_22_H_26_O_8_	418.4	100067
*G. corticate*	Bicyclo[3.2.1]oct-3-en-2-one, 3,8-dihydroxy-7-(7-methoxy-1,3-benzodioxol-5-yl)-6-methyl-5-(2-propenyl)-, [1S-(6-endo,7-exo,8-syn)]-	C_20_H_22_O_6_	358.39	101282028
	Cholesta-8,24-dien-3-ol, 4-methyl-, (3.beta.,4.alpha.)-	C_28_H_46_O	398.7	22212496
*P.boergesenii*	Glucobrassicin	C_16_H_20_N_2_O_9_S_2_	448.5	656506
	Glycitein	C_16_H_12_O_5_	284.26	5317750
	Matairesinol	C_20_H_22_O_6_	358.4	119205
	Naringenin	C_15_H_12_O_5_	272.25	932
	Pyrano [4,3-b] benzopyran-1,9-dione, 5amethoxy-9amethyl-3-(1-propenyl) perhydro	C_17_H_24_O_5_	308.4	5364482
	Syringaresinol	C_22_H_26_O_8_	418.4	100067
*S. wightii*	5-p-coumaroylquinic acid	C_16_H_18_O_8_	338.31	6441280
	Caffeic acid hexoside	C_15_H_18_O_9_	342.3	6124135
	Phloretin	C_15_H_14_O_5_	274.27	4788
	Quercetin-3-O-arabinoglucoside	C_26_H_28_O_16_	596.5	5484066

**Table 2 marinedrugs-20-00148-t002:** The binding affinity, RMSD, and interacting residues of the screened phytocompounds against RBD domain of omicron B.1.1.529. spike protein with the resolution of 2.79Å.

Seaweeds	Chemical Compound	Binding Affinity	RMSD (Å)	H/C-H Bond Interaction	Interaction Distances	Hydrophobic Interaction	Alkyl Interaction	Pi-Sigma /Cation Stacked Interaction
Standard drug	Ceftriaxone	−7.1	43.189	ARG403, ASN417, TYR453, SER494, SER496, TYR501	5.36, 4.21, 6.03, 3.24, 3.59, 6.23	ASP405, GLU406, ARG408, GLN409, LEU455, ARG493	-	ARG403, HIS505
	Cefuroxime	−5.3	2.483	THR376, GLY404 *, ARG408, TYR508	4.52, 5.06, 4.71, 5.71	PHE375, ASN437	ARG408, VAL503	-
*C. officinalis*	3’,8,8’-Trimethoxy-3-piperidin-1-yl2,2’-binaphthyl-1,1’,4,4’-tetrone	−6.9	1.897	-	-	ARG355, TYR396, ASP428, PHE429, THR430, SER514, PHE515, LEU517, LEU518	PRO426, PRO463	PHE464, GLU516
	Cholestan-3-ol, 2-methylene-, (3beta, 5 alpha)	−6.0	3.074	SER494	4.22	GLY482, THR470	LEU452, TYR449, ILE472, ALA484, PHE490	PHE490
*C. racemosa*	Glucobrassicin	−6.8	1.521	ARG457, ARG466, ASP467, ASP467 *	4.45, 6.50, 3.18, 5.49	ARG454, PHE456, SER459, GLU465, ILE468, SER469, TYR473, PRO491	ARG457, LYS458	ARG457, ASP467, GLU471
*G. corticata*	Bicyclo[3.2.1]oct-3-en-2-one, 3,8-dihydroxy-1-methoxy-7-(7-methoxy-1,3-benzodioxol-5-yl)-6-methyl-5	−6.3	20.322	THR430	4.16	ARG355, ASP428, SER514, PHE515, GLU516, LEU517	TYR396, PRO426, PHE429, PRO463, PHE464	-
	Cholesta-8,24-dien-3-ol, 4-methyl-, (3.beta.,4.alpha.)-	−6.8	1.544	-	-	TYR396, ASP428, THR430, GLU465, SER514, PHE515, GLU516	PRO426, LYS462, PRO463, PHE464	-
*P. boergesenii*	Glucobrassicin	−6.8	1.521	ARG454, LYS458, SER459, SER469	5.00, 4.44, 2.38, 3.25	PHE456, ARG457, TYR473, PRO491	-	ASP467, GLU471
	Glycitein	−6.1	11.668	THR376, ASP405 *	4.71, 4.5	GLY404, ARG408, VAL503, GLY504	VAL407	TYR508, PHE375
	Matairesinol	−6.0	1.707	ARG355, PHE464 *, SER514	6.65, 4.98, 3.72	TYR396, ASP428, PHE429, THR430, PRO463, PHE515, GLU516, LEU517	PRO426	-
	Naringenin	−6.4	20.209	ASN437, LYS440, LEU441	4.09, 4.44, 4.14	ASN343, PHE374, PHE375, SER438, ASN439	PRO373, LYS440	TRP436
	Pyrano [4,3-b] benzopyran-1,9-dione, 5a-methoxy-9a-methyl-3-(1-propenyl) perhydro	−6.1	29.617	SER496, TYR501	3.39, 5.24	ARG403, TYR453, TYR495, GLY502	HIS505	TYR501, HIS505
	Syringaresinol	−6.4	0.44	ARG355, ASP427 *, PRO463 *, PHE515 *, GLU516	6.88, 4.19, 4.81, 7.35, 3.78	ASP428, THR430, SER514	TYR396, PRO426, LYS462, PRO463	PHE464
*S. wightii*	5-p-coumaroylquinic acid	−6.0	4.96	ARG403, ASN417, TYR453, SER496	5.43, [2.96, 4.87], 5.62, 3.42	GLN409, GLY416, ILE418, LEU455, SER494, TYR495, TYR501, HIS505	-	-
	Caffeic acid hexoside	−6.4	2.82	ARG403, GLU406, ASN417, TYR453, SER496	[5.38, 6.24], 3.95, 4.96, 5.76, [3.34, 3.36]	ASP405, ARG408, GLN409, ILE418, LEU455, TYR495, SER494, TYR501, HIS505	-	-
	Phloretin	−6.3	0.061	TYR501, SER496, TYR453	5.37, 1.49, 5.90	ARG403, TYR495, PHE497, THR500, GLY502	-	TYR501, HIS505
	Quercetin-3-O-arabinoglucoside	−6.1	2.248	ASN331, THR333, GLY526, PRO527 *, LYS528C	5.09, [3.47, 3.38], [3.93, 4.29], 4.60, 5.10	PRO330, ILE332, CYS361, THR523	VAL362, CYS525	ASN360

Note: * = Carbon–hydrogen bond.

**Table 3 marinedrugs-20-00148-t003:** Drug-likeness and toxicity analysis for the selected compounds that inhibits the RBD of omicron B.1.1.529. spike protein.

Seaweeds	Chemical Compounds	Drug−Likeness						Toxicity Analysis							
		Mi LogP	TPSA	natoms	nON	nOHNH	*#*Violations	Intestinal Absorption	Hepato Toxicity	Carcino Genicity	Immuno toxicity	Muta Genicity	Cyto toxicity	LD50 (mg/kg)	TC
Standard Drug	Cefuroxime	−0.98	173.77	29	12	4	1	0.065	0.66^(Mild)^	0.50 ^(Mod)^	0.99^(−)^	0.76^(−)^	0.54^(Mod)^	10,000	VI
*C. officinalis*	3’,8,8’−Trimethoxy−3−piperidin−1−yl2,2’−binaphthyl−1,1’,4,4’−tetrone	3.99	99.22	36	8	0	0	0.606	0.83^(−)^	0.55^(Mod)^	0.70^(+)^	0.56^(Mod)^	0.58^(Mod)^	400	IV
	Cholestan−3−ol, 2−methylene−, (3beta, 5 alpha)	8.11	20.23	29	1	1	1	0.922	0.94^(−)^	0.62^(Mild)^	0.98^(+)^	0.94^(−)^	0.94^(−)^	5000	V
*G. corticate P.boergesenii*	Cholesta−8,24−dien−3−ol, 4−methyl−, (3.beta.,4.alpha.)−	7.96	20.23	29	1	1	1	0.931	0.82^(−)^	0.58^(Mod)^	0.97^(+)^	0.94^(−)^	0.96^(−)^	2000	IV
	Syringaresinol	2.62	95.86	30	8	2	0	0.599	0.87^(−)^	0.54^(Mod)^	0.95^(+)^	0.84^(−)^	0.99^(−)^	1500	IV
*S. wightii*	Caffeic acid hexoside	−0.77	156.91	24	9	6	1	0.221	0.82^(−)^	0.76^(−)^	0.95^(+)^	0.78^(−)^	0.87^(−)^	5000	V
	Phloretin	2.66	97.98	20	5	4	0	0.427	0.63^(Mod)^	0.72^(mild)^	0.98^(−)^	0.88^(−)^	0.82^(−)^	500	IV

Note: Mod = Moderate; − = Negative effect; + = Positive effect.

## Data Availability

The data presented in this study are available within the article.

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
