# Peer review of "In Silico Screening of Bioactive Compounds of Representative Seaweeds to Inhibit SARS-CoV-2 ACE2-Bound Omicron B.1.1.529 Spike Protein Trimer"

_marinedrugs, 2022, doi:10.3390/md20020148_

Round 1
Reviewer 1 Report
This is a very innovative in silico study reporting on seaweed-derived compounds as poteintial antiviral candidates, with special interest on SARAS-CoV-2 variants.
I have only minor corrections to point out:
- Starting on line 180, some residues are marked with asterisk (*). The meaning of this marking is not stated in the text (is it the same as the asterisk in Table 2: - C-H bonds?)
- line 57: change 'infectious' to 'infectiousness'
- line 61: change 'Manaus ' to 'Brazil'
- line 67: change 'indicating' to 'indicating more'
- lines 100-101, 304, 313, 315, 322: change reference style to numbers, check if listed at the end of the manuscript
- line 377: change 'founded' to 'found'
Author Response
Reviewer: 1
- Comment 1: Starting on line 180, some residues are marked with asterisk (*). The meaning of this marking is not stated in the text (is it the same as the asterisk in Table 2: - C-H bonds?)
Author response: Thank you very much for all the kind comments and suggestions. Yes, the meaning for * is C-H bonds. We have mentioned the term instead of asterisk. Kindly verify Page No. 5 and Line No. 182-183 in the revised manuscript.
- Comment 2: line 57: change 'infectious' to 'infectiousness'
Author response: As per the reviewer’s comment we have changed to ‘infectiousness’. Kindly verify Page No. 2. Line No. 58.
- Comment 3: line 61: change 'Manaus ' to 'Brazil'
Author response: As per the reviewer’s comment we change to Brazil. Kindly verify Page No. 2 Line No. 61.
- Comment 4: line 67: change 'indicating' to 'indicating more'
Author response: As per the reviewer’s comment we have changed to ‘indicating more’. Kindly verify Page No.2. Line No. 67.
- Comment 5: lines 100-101, 304, 313, 315, 322: change reference style to numbers, check if listed at the end of the manuscript.
Author response: As per the reviewer’s comment, we have changed the reference style, assigned the number, and modified the remaining reference number throughout the manuscript. Kindly verify Page No. 3. Line No. 100-101, Page No. 4. Line No. 311-313; 319-322. Also, we added the missed references which were cited in the manuscript as a text previously. Kindly verify Page No. 8 Line No. 524; Page No. 9. Line No. 575-590.
- Comment 6: line 377: change 'founded' to 'found'
Author response: As per the reviewer’s comment, we have changed to ‘found’. Kindly verify Page No.5. Line No. 388.
Reviewer 2 Report
The manuscript by Bharathi et al. describes a docking study of 96 compounds isolated from seaweeds regarding their potential to inhibit SARS-CoV-2 ACE2-bound Omicron 3 B.1.1.529 Spike Protein. The most perspective compounds were selected, their binding and ADMET characteristics were discussed. It was shown that the seaweeds components are promising compounds for development of antiviral drugs and food supplements. The results are important and deserve publication.
Specific comments:
Section 2.1:
All the 96 compounds under investigation should be listed in Supplementary.
Lines 126-127:
The sentence is grammatically incorrect. Please check.
Lines 132-133:
There is a strange phrase "docking and docked".
Line 140:
"distance" should be replaced with "distances".
Lines 140-141:
The sentence "∑ is the overal pairs of atoms" is mathematically and grammatically incorrect.
Line 181:
There is a strange term "pi-alky bond".
Line 189:
It would be more clear to give the information concerning the toxicity class in a separate sentence.
Table 2:
The term "bond length" is incorrect with respect to the presented interactions. It should be replaced with the "interaction distance".
Table 2:
Since RMSD is a relative measure, the reference structure for RMSD calculation should be clearly indicated.
Table 3:
Some notations listed under "Drug-likeness" were not mentioned in Section 2.4.
Caption to Figure 2:
The uncommon term "docking posture" should be replaced with the more widely used term "docking pose". Panels B, E, H, K, N, and Q depict the receptor surface colored according to hydrophobic interactions. However, the word "surface" is not used in the figure caption.
Lines 285-292:
Again, a reference structure for RMSD calculation should be mentioned.
Line 306:
Please replace "hydrogen bond interactions" with "hydrogen bonding interactions".
Summarizing, I recommend acceptance of the manuscript for publication after minor revision.
Author Response
- Comment 1: Section 2.1: All the 96 compounds under investigation should be listed in Supplementary.
Author response: Thank you very much for all the kind comments and suggestions. As the reviewer said, we have created the supplementary Table S1 and included all the 96 compounds and their seaweeds, binding affinity, RMSD, and references. Also, mentioned the supplementary table in the revised manuscript in section 2.1 and the result section. Kindly verify Page No. 3 Line No.109-110. Page No.4 Line No. 166 in the revised manuscript and verify Table S1 in the revised supplementary file 1.
- Comment 2: Lines 126-127: The sentence is grammatically incorrect. Please check.
Author response: As per the reviewer’s comments, we have corrected the sentence grammatically and meaningly. Kindly verify Page No.3 Line No. 126-128.
- Comment 3: Lines 132-133: There is a strange phrase "docking and docked".
Author response: As per the reviewer’s comment, we have changed the sentence completely. Kindly verify Page No.3 Line No. 132-133.
- Comment 4: Line 140: "distance" should be replaced with "distances".
Author response: As per the reviewer’s comment, we have changed the word to “distances”. Kindly verify Page No. 3 Line No. 140.
- Comment 5: Lines 140-141: The sentence "∑ is the overall pairs of atoms" is mathematically and grammatically incorrect.
Author response: As per the reviewer’s comment, we have corrected the summation according to the mathematical meaning. Kindly verify Page No. 3. Line No. 140-141.
- Comment 6: Line 181: There is a strange term "pi-alky bond".
Author response: As per the reviewer’s comment, we have corrected the term as “pi-alkyl” bond in the sentence. Kindly verify Page No. 5. Line No. 181-182.
- Comment 7: Line 189: It would be more clear to give the information concerning the toxicity class in a separate sentence.
Author response: As per the reviewer’s comments, we have mentioned toxicity analysis in separate sentences and described it in the discussion section by adding a few references. Page No. 5. Line No. 1186, 191-192; Page No. 6. Line No. 197-198. Page No. 10. Line No. 209-210, 213-214, 217. Page No.17. Line No. 358-363 in the discussion section and reference section Reference No. 76, 77. Line No. 602-605.
- Comment 8: Table 2: The term "bond length" is incorrect with respect to the presented interactions. It should be replaced with the "interaction distance".
Author response: As per the reviewer’s comment, we have changed the term into “interaction distances”. Kindly verify Page No. 7. Table 2.
- Comment 9: Table 2: Since RMSD is a relative measure, the reference structure for RMSD calculation should be clearly indicated.
Author response: As it is the PDB 7T9J structure without ligand we could only able to mention the protein structure resolution. Kindly verify, Page No.7. Line No. 202-203.
- Comment 10: Table 3: Some notations listed under "Drug-likeness" were not mentioned in Section 2.4.
Author response: As per the reviewer’s comment we have included the missed ADMET properties in the materials and methods section 2.4. Kindly verify Page No. 3 Line No. 147-148.
- Comment 11: Caption to Figure 2: The uncommon term "docking posture" should be replaced with the more widely used term "docking pose". Panels B, E, H, K, N, and Q depict the receptor surface colored according to hydrophobic interactions. However, the word "surface" is not used in the figure caption.
Author response: As per the reviewer’s comment, we have changed the term into "docking pose". Also, we have added the term surface in the figure caption. Kindly verify Page No. 13. Line No. 247-248.
- Comment 12: Lines 285-292: Again, a reference structure for RMSD calculation should be mentioned.
Author response: As it is the PDB 7T9J structure without ligand we could only able to mention the protein structure resolution. Kindly verify, Page No. 16. Line No. 299-300.
- Comment 13: Line 306: Please replace "hydrogen bond interactions" with "hydrogen bonding interactions".
Author response: As per the reviewer’s comments, we have replaced the term with "hydrogen bonding interactions". Kindly verify Page No. 5. Line No. 180; Page No. 16. Line No. 314, Page No. 17. Line No. 349.